# Severity of Placental Abruption in Restrained Pregnant Vehicle Drivers: Correct Seat Belt Use Confirmed by Finite Element Model Analysis

**DOI:** 10.3390/ijerph192113905

**Published:** 2022-10-26

**Authors:** Katsunori Tanaka, Yasuki Motozawa, Kentaro Takahashi, Tetsuo Maki, Mami Nakamura, Masahito Hitosugi

**Affiliations:** 1Department of Legal Medicine, Shiga University of Medical Science, Otsu 520-2192, Japan; 2Hino Memorial Hospital, Hino 529-1642, Japan; 3Department of Mechanical Engineering, Tokyo City University, Tokyo 158-8557, Japan

**Keywords:** pregnant women, motor vehicle collision, placental abruption, numerical simulation, finite element model, seat belt, safety equipment, anterior superior iliac spine, sled test

## Abstract

Despite wearing a seat belt, pregnant drivers often suffer from negative fetal outcomes in the event of motor accidents. In order to maintain the safety of pregnant drivers and their fetuses, we assessed the severity of placental abruption caused by motor vehicle collisions using computer simulations. We employed a validated pregnant finite element model to determine the area of placental abruption. We investigated frontal vehicle collisions with a speed of 40 km/h or less involving restrained pregnant drivers with a gestational age of 30 weeks. For a crash speed of 40 km/h, the placental abruption area was 7.0% with a correctly positioned lap belt across the lower abdomen; it was 36.3% with the belt positioned at the umbilicus. The area of placental abruption depended on collision speed, but we found that with a correctly positioned belt it likely would not lead to negative fetal outcomes. We examined the effects on placental abruptions of reconfiguring seat belt width and force limiter setting. A wider lap belt and lower force limiter setting reduced the area of placental abruption to 3.5% and 1.1%, respectively; however, they allowed more forward movement upon collision. A 2.5 kN force limiter setting may be appropriate with respect to both forward movement and reduced placental abruption area. This study confirmed the importance of correctly using seat belts for pregnant drivers. It provides valuable evidence about improving safety equipment settings.

## 1. Introduction

Approximately 1.3 million people die annually in motor vehicle collisions (MVCs); they are the leading cause of death for children and young adults aged 5–29 years [1]. One study based on national MVC data in the United States found that the odds for a belt-restrained female driver to sustain severe injuries were 47% higher than for a belt-restrained male driver involved in a comparable crash [2]. Therefore, preventing female drivers of reproductive age from fatal MVCs is a high priority worldwide. Among pregnant women, MVCs are the leading cause of fetal death related to maternal trauma [3]. Approximately 3% of pregnant women are involved in MVCs [4,5]. Maintaining both maternal and fetal health requires minimizing the risk of harm from MVCs.

According to one study about fetal loss following maternal involvement in MVCs, placental abruption was the major cause; it accounted for 50–70% of negative fetal outcomes [6]. In MVCs, great forces are applied to the uterus via the abdomen, and shearing forces along the uteroplacental interface (UPI) occur. When considering the severity of placental abruption, it is necessary to focus on the area of uteroplacental separation. If a placental abruption occurs with only a small separated area following the abdominal trauma, the pregnant woman may suffer no adverse changes and no symptoms. One study examined separated areas of the UPI and the proportion of fetal fatalities. It found that separated areas of under 30%, 30–50%, and over 50% corresponded, respectively, to fetal fatalities after vaginal delivery of 30%, 79%, and 100% [7].

In order to decrease the separated area, we previously examined the factors that promote placental abruptions in MVCs using a finite element (FE) model. We developed a novel FE model for a small pregnant woman with a gestational age of 30 weeks sitting in the driver seat [8]. The model could quantitatively evaluate the separated area in placental abruption in a simulated abdominal injury caused by pressure from the lower rim of the steering wheel [8]. In that simulation, the crash speed and the steering wheel position were key factors in the severity of placental abruption. Even at a low collision speed, a negative fetal outcome could occur if a seat belt was not worn, owing to a large placental abruption. We quantitatively analyzed the area of placental abruption in a numerical simulation of an unrestrained pregnant vehicle driver at collision velocities of 10.8 and 21.6 km/h. We found that over that collision speed range, the area of placental abruption increased 10-fold and accounted for approximately 20% of the total abruption area, owing to pressure from the steering wheel. We concluded that reduced collision speed and seat belt restraints are important for decreasing the area of placental abruption.

Most pregnant drivers now wear seat belts. One survey in Japan revealed that 94.8% of pregnant drivers in 2009 always used a seat belt [9]. However, pregnant women often suffer from negative fetal outcomes despite wearing a seat belt. One study based on real-world MVCs reported that approximately 83% of restrained pregnant drivers suffered negative fetal outcomes when involved in a frontal collision with a crash speed of 48 km/h or more [10]. This finding may have been due in part to inappropriate seat belt use. According to one investigation that examined the method of seat belt use, 13.1% of pregnant drivers who always wore a seat belt did so incorrectly [11]. Among those incorrect uses, the most common was having the lap belt cross over the abdomen [11].

Placental abruptions can occur for restrained pregnant drivers, so we examined the factors that lead to such abruptions in collisions. In this study, we first confirmed the effects of lap belt position and collision speed on the uteroplacental separation area; second, with the goal of developing future seat belt systems, we examined changes in the width of the lap belt and the activating conditions of the force limiter.

## 2. Materials and Methods

We considered placental abruptions following a frontal MVC involving a small pregnant woman with late gestational age. The flowchart of our study is illustrated in Figure 1. Small pregnant women are common in Asia (including Japan), and late-term pregnant women are believed to experience greater abdominal trauma owing to abdominal protrusion. According to studies using a national collision database (NASS-CDS), frontal collisions are the most frequent direction of collision, accounting for nearly 50% [12,13]. Despite the various types of MVC and analyses of risk factors [14,15,16], we focused on simulation of frontal collisions. We used an FE model of pregnant drivers that had been constructed and validated to determine the area of placental abruption [8]. The FE method is an approximate solution of partial differential equations; currently, it is widely used in various scientific and technological fields, including automotive engineering [17,18,19,20].

To analyze MVCs close to a real-world setting, we constructed and validated a sled impact test FE model (detailed below). In this FE model, we installed commonly utilized safety equipment devices, including the air bag and seat belt.

### 2.1. FE Model of Pregnant Woman

On the basis of the above assumptions, we set the height and gestational age for the pregnant FE model at 153 cm and 30 weeks. The height corresponds to the fifth percentile among American women but is more common in Asia. The details are described in another study [8].

### 2.2. FE Model of Driver’s Seat

We constructed a vehicle FE model, comprising a driver seat, three-point seat belt, air bag, steering wheel, knee bolster, and the floor (Figure 2). The three-point seat belt consisted of the shoulder and lap belt. We based the model on the vehicle FE models published by National Highway Traffic Safety Administration in the United States. As medium-sized sedans that are commonly driven worldwide, we chose the Ford Taurus for the driver seat FE model, the Honda Accord for the seat belt and air bag FE models, and the Toyota Yaris for the other FE models.

We installed the seat belt FE model with a pretensioner and force limiter, both of which enhance safety. To prevent a driver or passenger from sliding forward on the seat, a pretensioner retracts the seat belt quickly in the event of a crash. The person is thus tightly restrained to avoid collision with the steering wheel or dashboard. However, the force exerted by the seat belt can cause injury. A force limiter thus allows extension of the seat belt over a short length to reduce the restraint and avoid injury when the load reaches a configured value.

There were 167,352 nodes and 266,233 elements in the FE model for the sled impact test simulation, including the pregnant FE model. In constructing the FE model, we used a commercial modeling tool, Hyperworks, by Altair (Altair Engineering Inc., Troy, MI, USA).

### 2.3. Sled Impact Test Simulation

We conducted a numerical simulation using the developed sled impact test FE model. For a frontal collision, we applied crash acceleration to the FE model. Air bag inflation and the pretensioner functioned in all study simulations. The force limiter also functioned using the settings described below.

#### 2.3.1. Placental Abruption Area Analysis in Typical Safety Equipment Settings

To analyze the placental abruption area with a typical safety equipment setting, we first ran numerical simulations in which we varied the crash speed and lap belt position. For three velocities of 10, 25, and 40 km/h, we accelerated the sled impact test FE model with the waveforms presented in Figure 3, which were measured in real sled impact tests using a pregnant dummy, MAMA-2B [21,22,23]. We ran the simulations for three lap belt positions for the upper, middle, and lower abdomen as illustrated in Figure 4a–c, respectively. The lap belt for the lower abdomen was positioned to hold the anterior superior iliac spines bilaterally, which is recommended as the correct position for general adults for effective body restraint [24]. The lap belt for the middle and upper abdomen was positioned, respectively, at the umbilicus and 30 mm higher (which corresponds to the height of the uterine fundus). The seat belt width was 50 mm, and the force limiter was configured to 2.5 kN.

The separated areas were expressed as percentages for the whole UPI. The numerical simulations computed stresses for all similarly sized finite elements and the UPI, as illustrated in Figure 5a,b. We considered UPI elements that exceeded the breaking stress (15.6 kPa, taken from the literature [25]) to have separated. UPI elements considered to be separated are colored red in Figure 5b. For all UPI elements, we determined the separated areas as proportions of the separated UPI elements.

#### 2.3.2. Mitigating Placental Abruption by Reconfiguring Safety Equipment

To achieve better safety equipment settings, we ran simulations in which we varied the lap belt widths and force limiter settings. For the three velocities, we ran the simulations for three lap belt widths of 50, 75, and 100 mm with the 2.5 kN force limiter of the lap belt. We ran the simulations for three force limiter settings (2.5, 2, and 1 kN) for the 50 mm lap belt.

## 3. Results

We validated the sled impact test FE model by comparing it with the real sled impact test using the pregnant dummy, MAMA-2B. With our validated FE model, we performed an initial analysis of placental abruption with typical safety equipment settings. For smaller separated areas, we reconfigured the safety equipment settings and undertook secondary analysis of the placental abruption.

### 3.1. Validation of Sled Impact Test FE Model

To validate the results of this study, we fitted two acceleration waveforms for the chest and lumbar regions between the simulation and the measurement in the real sled impact tests using the pregnant dummy, MAMA-2B. Both the simulation and real measurement assumed frontal collision at 40 km/h. In the simulation, we applied the acceleration waveform at 40 km/h (illustrated in Figure 3) to obtain the chest and lumbar acceleration waveforms.

The results appear in Figure 6 and Figure 7. The acceleration waveforms were mostly isomorphic; the peak acceleration of the chest and lumbar regions in the simulation varied, respectively, just by +4.8% and +2.7% from the values in the real sled impact test. We thus concluded that our current FE model was dynamically valid.

### 3.2. Determining Placental Abruption Areas with Typical Safety Equipment Settings

Figure 8 depicts the placental abruption areas for crash velocities of 10, 25, and 40 km/h and for lap belt positions for the lower, middle, and upper abdomen. At 10 km/h, the abruption areas were 0.5%, 2.0%, and 1.6%, respectively, for the lower, middle, and upper abdominal lap belt positions. At 25 km/h, the abruption areas were 4.4%, 27.3%, and 16.7%, respectively, and at 40 km/h they were 7.0%, 36.3%, and 26.3%.

### 3.3. Determining Placental Abruption Areas with Reconfigured Safety Equipment

Figure 9 presents the placental abruption areas at 40 km/h for lap belt widths of 50, 75, and 100 mm. For 50 mm, the placental abruption areas were 6.1, 36.3, and 26.3%, respectively, for the lower, middle, and upper abdominal lap belt positions. For 75 mm, the areas were 4.2, 24.3, and 16.7%, respectively; for 100 mm, the areas were 3.5, 18.1, and 14.5%, respectively.

Figure 10 illustrates the placental abruption areas at 40 km/h for the force limiter settings of 2.5, 2, and 1 kN. With the force limiter at 2.5 kN, the placental abruption areas were 6.1, 36.3, and 26.3%, respectively, for the lower, middle, and upper abdominal lap belt positions. With the 2 kN setting, the areas were 3.7, 14.2, and 12.8%, respectively; with 1 kN, the areas were 1.1, 13.7, and 10.9%, respectively.

Lower force limiter settings allow the driver to move forward in the seat. Figure 11 shows the forward movement of the pregnant driver in the FE model with different force limiter settings. When the force limiter for the lap belt was 2.5 kN, the forward movement was 79.6, 81.2, and 95.1 mm, respectively, for lap belt positions for the lower, middle, and upper abdomen at 40 km/h. For 2 kN, the values were 120.3, 133.1, and 148.9 mm, respectively; for 1 kN, they were 127.3, 137.1, and 152.2 mm, respectively.

## 4. Discussion

Using data from both pregnant women and sled impact test FE models, we analyzed the placental abruption area resulting from frontal collisions with seat belt restraint and air bag deployment. First, we confirmed the importance of correct seat belt use by means of simulations with varying lap belt positions.

According to the simulation results in Figure 8, when the lap belt was positioned incorrectly (i.e., on the upper or middle abdomen), the abruption area increased considerably. For example, when the lap belt was positioned incorrectly, the abruption area at 25 km/h increased by more than 10-fold over the area at 10 km/h, and at 40 km/h it increase by over 16-fold. These results are in accordance with those of previous studies of pregnant drivers not wearing a seat belt involved in frontal collisions [8]. The rear-to-front vehicle distance is an important factor, as well as vehicle speed. When the distance is short and the preceding vehicle stops suddenly, even if the succeeding vehicle brakes as quickly as possible, the succeeding vehicle speed at collision is estimated to still be high, and the high-speed collision increases the area of placental abruption. Therefore, if the seat belt position is inappropriate, negative outcomes can occur similar to those with unbelted driving. We found that if the lap belt was used properly, the abruption area increased according to the collision speed. We observed that the separated area was just 7.0% at 40 km/h, which would be expected to exert no or a small negative impact on fetal outcomes. The risk of placental abruption decreases with lower collision speed; thus, it is necessary to promote education about maintaining vehicle velocities within the regulated limit—especially among pregnant drivers.

Second, we varied the safety equipment settings (lap belt width and force limiter setting) to evaluate the effect of reconfiguration. We observed that a wider lap belt could help mitigate placental abruption. With a correctly positioned lap belt, if the belt width was doubled from 50 to 100 mm, the separated area decreased from 6.1% to 3.5%. Even when incorrectly positioned, the wider belt halved the separated area—though the abruption was still severe. Because the force was distributed to a wider belt area, the applied pressure decreased. Thus, according to our results, wider seat belts could improve both maternal and fetal outcomes. However, as shown in previous results, some pregnant women hesitate to put the lap belt in contact with the abdomen due to discomfort [26,27]. Wider belts could positively affect seat belt use. Further research is required to investigate better seat belt widths for pregnant passengers and drivers that can promote both high seat belt use and low applied pressure.

A lower force limiter setting could reduce the abrupted area. The abrupted area was 6.1% for a 2.5 kN force limiter when the lap belt was correctly positioned, but it was just 1.1% at 1 kN. However, if the seating position of a pregnant driver is close to the steering wheel, a lower setting of the force limiter could bring the abdomen into contact with the wheel. With our model, though, we found that the lower rim of the steering wheel did not strike the abdomen because the inflated air bag kept the upper body of the pregnant driver away from the steering wheel and did not apply to the abdomen (Figure 12). This result coincides with observations reported in the literature [28] and implies that coordination among safety devices and their appropriate settings could prevent placental abruption and improve maternal and fetal outcomes.

One study of the seating position and anthropometric parameters of pregnant Japanese drivers at around 31 weeks’ gestation found that the mean horizontal clearance between the lower rim of the steering wheel and the driver’s abdomen was 146 mm [29]. That study also reported that the drivers’ height and the horizontal distance from the lower rim of the steering wheel to the abdomen showed a moderate linear correlation. With a correctly positioned seat belt, we found that the simulated forward movement with a 1 kN force limiter setting was within the distance between the steering wheel and the abdomen of average Japanese pregnant women. However, the seating position, degree of protrusion of the abdomen, and collision speed would influence contact between the abdomen and steering wheel. With a 2.5 kN force limiter setting at 40 km/h, the abruption area was just 6.1% and forward movement was limited to less than 80 mm. Even such conditions might, however, result in contact between the steering wheel and abdomen at higher collision speeds for smaller pregnant women or for women of later gestational age. Accordingly, we consider the 2.5 kN force limiter setting appropriate. Further research from both medical and mechanical perspectives is necessary to identify the ideal force limiter setting to ensure safety.

The present study has several limitations. First, in this investigation, the placenta was located only at the uterine fundus, not at the frontal, rear, or lateral uterine wall. The interaction between the forces via the lap belt and area of placental abruption could depend on the position of the placenta. In the future, more simulations are required to vary the placental position. Second, in this study, we constructed only one size for the pregnant FE model—153 cm height and 54 kg weight. The abdominal force from the lap belt depends on the body weight of the pregnant driver. Therefore, our results could have differed with a different body weight. Third, the placental region covered by the lap belt depends on the body height. For comparison, a pregnant woman FE model with different body sizes should be employed in future. Fourth, we simulated only frontal crashes and only at 40 km/h or lower. Other collision directions and higher collision speeds may induce more negative fetal outcomes; furthermore, simulations are required to predict injury severities for both mothers and fetuses.

## 5. Conclusions

This paper simulated placental abruption following a frontal MVC using an FE model of a pregnant woman:According to our results, a correctly positioned lap belt could prevent negative fetal outcomes by minimizing the area of placental abruption to 7.0% at 40 km/h.Our findings also suggest that a wider lap belt could mitigate placental abruption. A force limiter setting of 2.5 kN may be adequate with respect to both forward movement and placental abruption area.We confirmed the safety factors related to correct seat belt use among pregnant drivers.

This study provides valuable evidence to improve safety equipment settings. Analysis of higher speed collisions over 40 km/h, various types of collision, and various positions of the placenta among pregnant women might provide further evidence to improve safety for both pregnant women and their fetuses. Machine learning can help in analyzing real-world MVCs. Incorporation of risk factors identified in machine learning-based analyses with analysis of placental abruption based on computer simulation would contribute to further improving safety.

## Figures and Tables

**Figure 1 ijerph-19-13905-f001:**
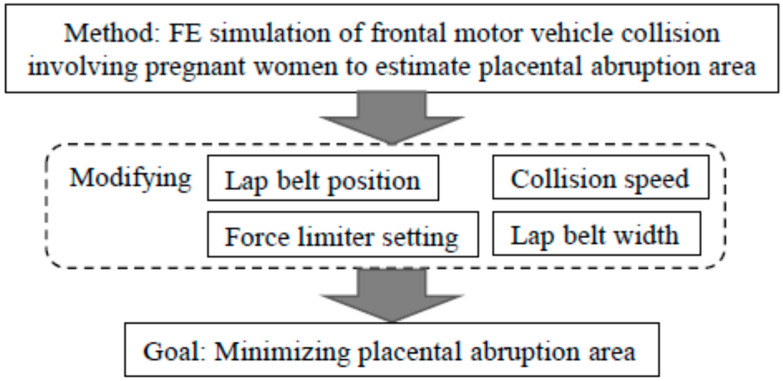
Flowchart of the study.

**Figure 2 ijerph-19-13905-f002:**
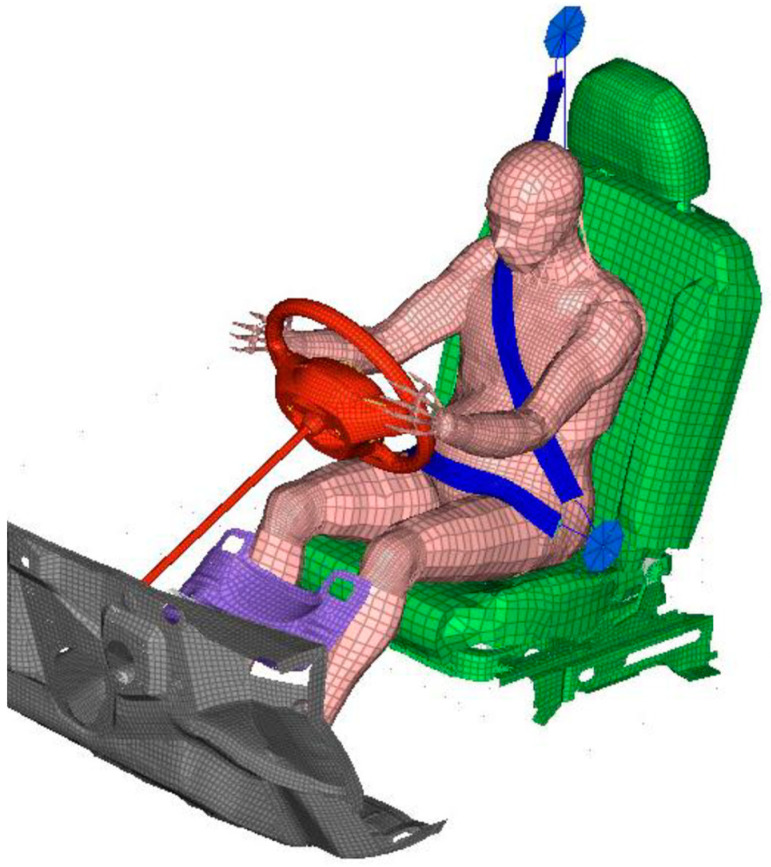
FE model of sled impact test environment.

**Figure 3 ijerph-19-13905-f003:**
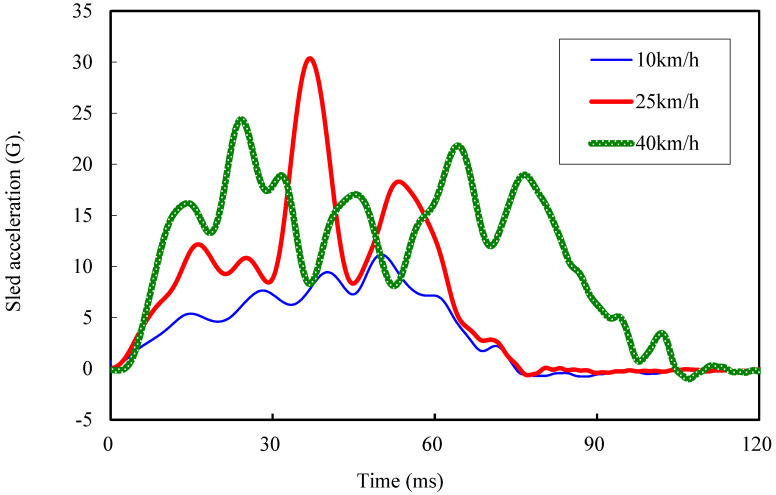
Sled acceleration.

**Figure 4 ijerph-19-13905-f004:**
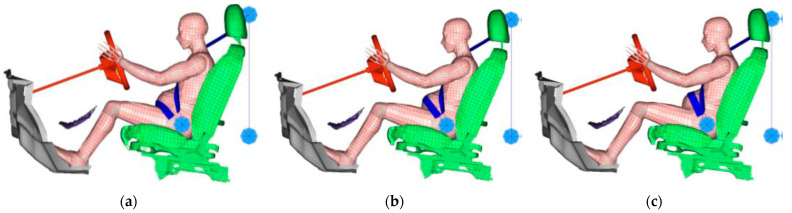
Lap belt positions: (**a**) lower abdomen (correct position), (**b**) middle abdomen, and (**c**) upper abdomen.

**Figure 5 ijerph-19-13905-f005:**
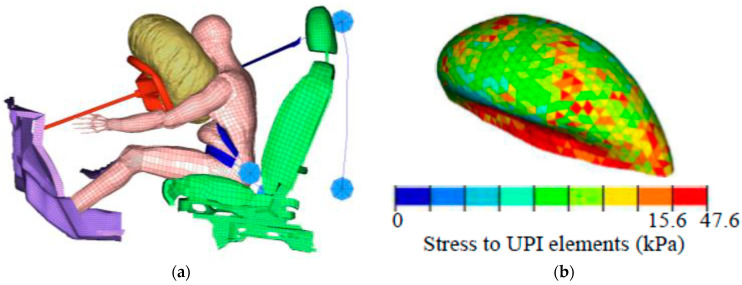
(**a**) Finite element model of pregnant drivers in sled test simulation at 90 ms after collision at 40 km/h, and (**b**) Uteroplacental interface (UPI) elements colored according to computed stress. Red UPI elements exceed the breaking stress, 15.6 kPa.

**Figure 6 ijerph-19-13905-f006:**
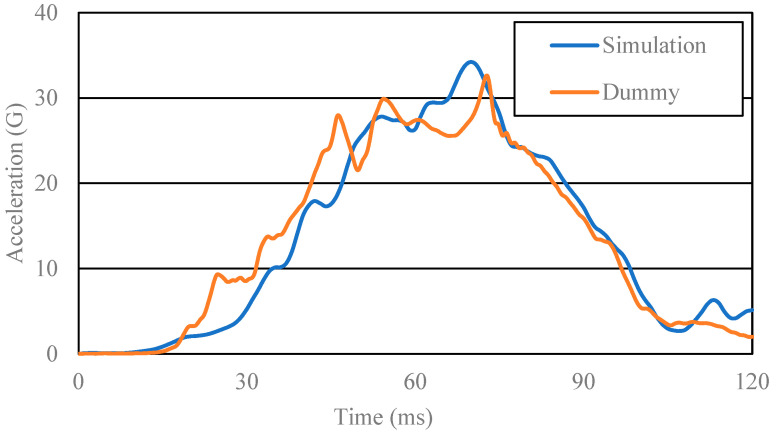
Chest acceleration waveforms from the simulation and from the real sled impact test using the pregnant dummy.

**Figure 7 ijerph-19-13905-f007:**
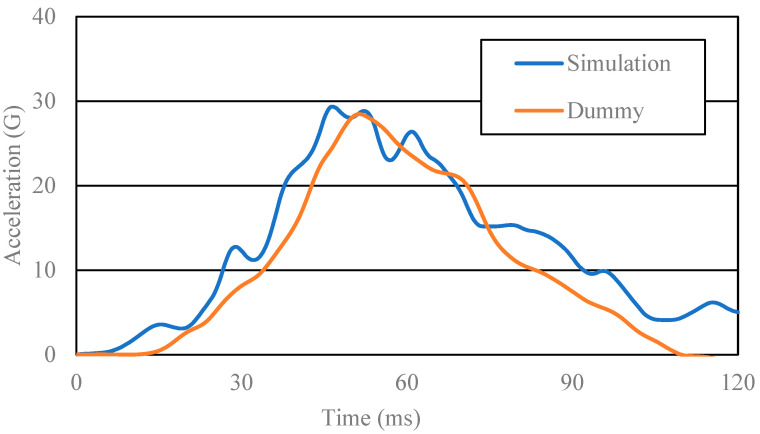
Lumbar acceleration waveforms resulting from the simulation and from the real sled impact test using the pregnant dummy.

**Figure 8 ijerph-19-13905-f008:**
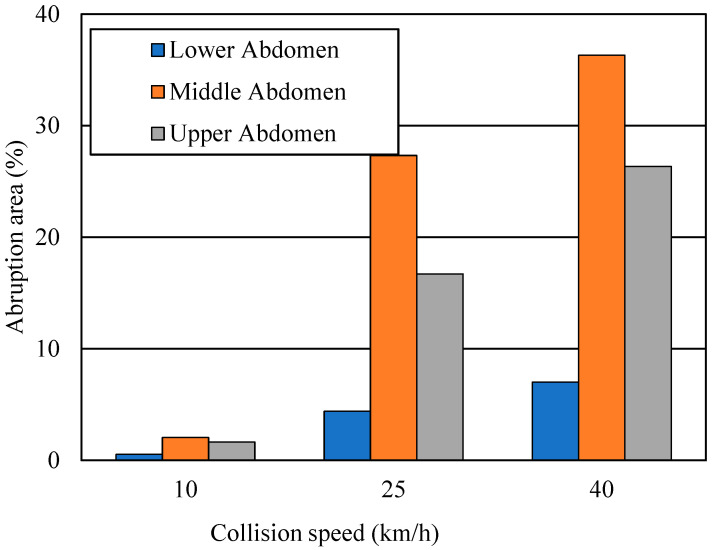
Placental abruption areas according to crash velocities and lap belt positions.

**Figure 9 ijerph-19-13905-f009:**
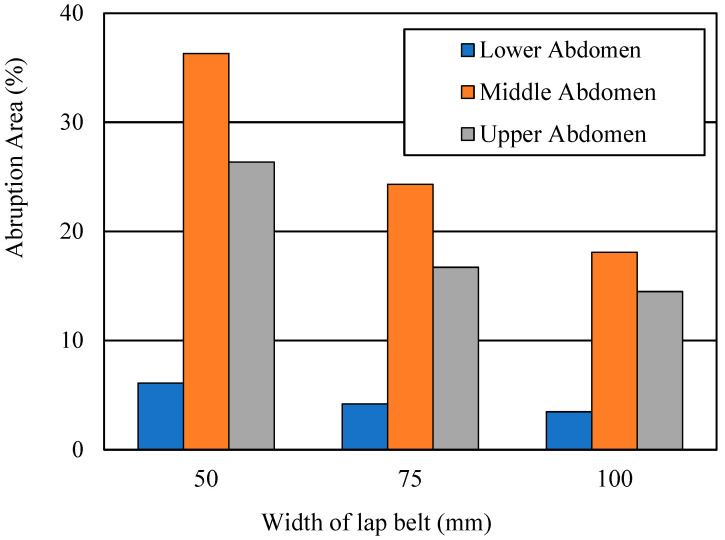
Placental abruption areas for lap belt widths and positions at 40 km/h.

**Figure 10 ijerph-19-13905-f010:**
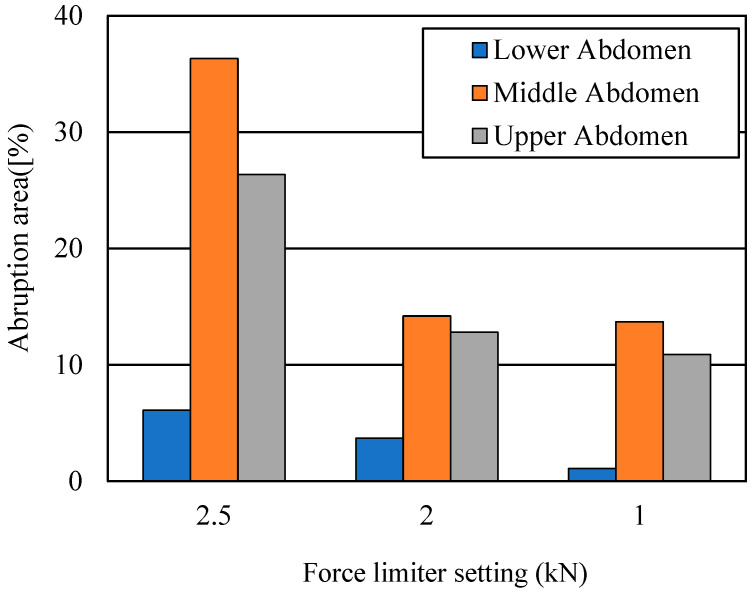
Placental abruption areas for force limiter settings and lap belt positions at 40 km/h.

**Figure 11 ijerph-19-13905-f011:**
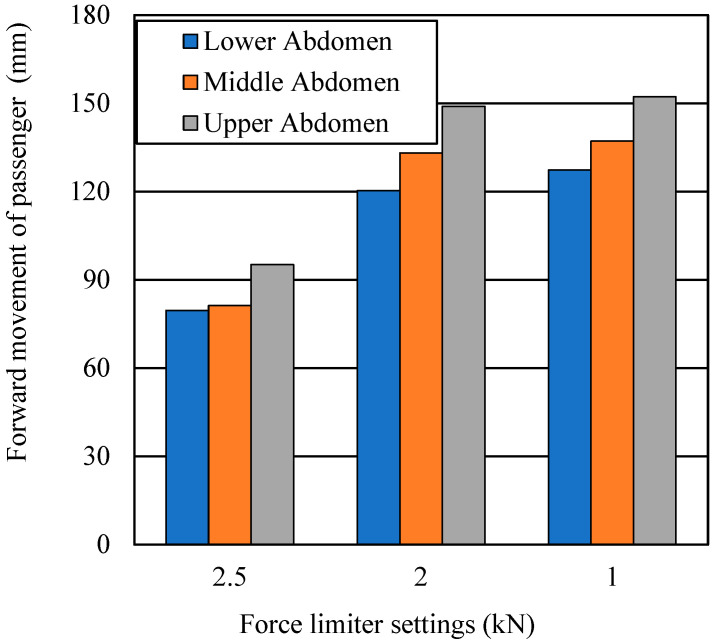
Forward movement of pregnant driver with different force limiter settings.

**Figure 12 ijerph-19-13905-f012:**
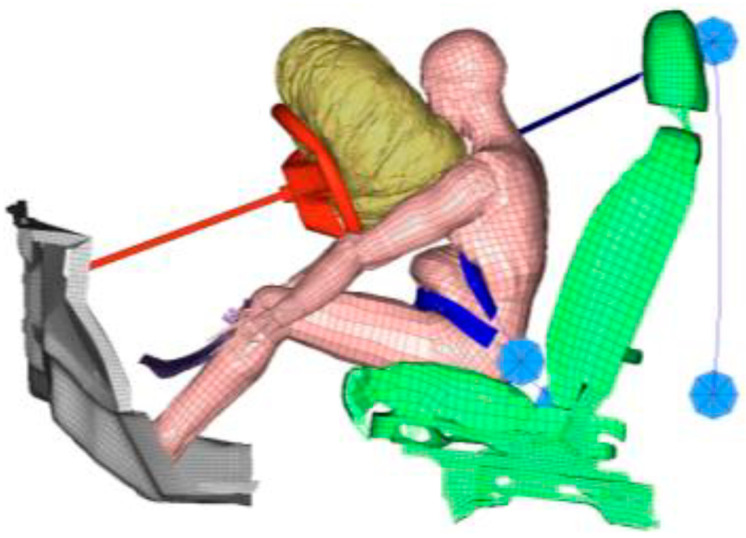
Preventing steering wheel pressure with an inflated air bag with a 1 kN force limiter for a 50 mm lap belt positioned across the lower abdomen.

## Data Availability

The data presented in this study are available upon request from the corresponding author.

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
