# Peer review of "Severity of Placental Abruption in Restrained Pregnant Vehicle Drivers: Correct Seat Belt Use Confirmed by Finite Element Model Analysis"

_ijerph, 2022, doi:10.3390/ijerph192113905_

Round 1

Reviewer 1 Report

The research's quality is acceptable for publication with minor corrections. The suggested modifications are writen below.

- An illustration of the UPI failure elements should be necessary to provide an example about how the abruption area is calculated. What is the breaking stress considered and the element size implemented?

- Maybe the third and fourth sections should be merged. Sections 3.1, 3.2 and 3.3 need a deeper discussion of the results. They only describe the results, without explaining their implications.

- An image of the abdominal lap belt positions would clarify the comprehension of this studied variable.

Reviewer 2 Report

The authors used an effective pregnancy FEM model to determine the area of placental abruption, studying the effects of pregnant drivers at 30 weeks of pregnancy, and considered the width of the seat belt, the different position of the seat belt, and the effects of different limiters on placental abruption. Finally, the area of placenta was compared to determine the influence of each variable factor.

This paper is different from the traditional occupant injury research idea point, unique ideas, but in some places, the analysis and expression is not clear enough, need to be clarified and simplified, the paper needs to be greatly revised. My specific comments are as follows:

1What is the basis of choosing the finite element manikin with a gestational age of 30 weeks?

2How is the curve filtering of Figure 2,3 and 4 handled?

3Line 59“We quantitatively analyzed the area of placental abruption in a numerical simulation of an unrestrained pregnant vehicle driver at collision velocities of 10.8 and 21.6km/h.”Numerical simulation of 10.8km/h and 21.6km/h collision speed for unconstrained system pregnant vehicle drivers. Where are the simulation results? The latter text is not mentioned.

4Line 102-104“As medium sized sedans commonly driven worldwide, we chose the Ford Taurus for the driver seat FE model, Honda Accord for the seat belt and air bag FE models, and Toyota Yaris for the other FE models.”Why is the combination of different brands of vehicle components into a finite element model used? What is the difference between the combined vehicle model and a single brand vehicle model?

5Line 118“For a frontal collision, we applied crash acceleration to the FE model.”How is the collision acceleration obtained here?

6“3.1Validation of sled impact test FE model”Real vehicle crash test and finite element model crash test comparison verification model effectiveness, crash test conditions are not stated?

7What is the simulation test collision speed (10 km/h, 25 km/h, 40 km/h) selected based on?

8What is the basis of the seat belt force limit value setting?

9Line 218“With a correctly positioned lap belt, if the belt width was doubled from 50 to 100 mm, the separated area decreased from 7.0% to 3.5%.”At a collision speed of 40 km/h, with a seat belt width of 50mm, the placental abruption area was 6.1%, not the 7.0% as described here.

10Line 236“With our model, we found, though, that the lower rim of the steering wheel did not strike the abdomen because the inflated air bag kept the upper body of the pregnant driver away from the steering wheel . ”Due to the presence of the airbag, the steering wheel has no contact with the abdomen, is it necessary to consider the impact of the airbag on placental abruption here?

Reviewer 3 Report

This paper is wonderful and it has novelty for publishing in your journal. However, in my opinion, the manuscript has some shortcomings in the text. According to the mentioned items, I recommend minor revisions for the manuscript.

·       Why authors haven’t investigated other collision directions? In many papers, all types of collision have been considered for accident severity modeling. Therefore, the authors should describe the reason for using just one collision direction by citing the following papers:

            https://doi.org/10.1007/s42452-020-04081-3

            https://doi.org/10.1016/j.aap.2010.01.001

            https://doi.org/10.1155/2021/9974219

·       In whole text of paper, you should change the word “velocity” with “speed”. In Safety papers, speed is more common than velocity.

·       In discussion section, the authors have mentioned correctly considering the limitation speed as a one of the main reason to decrease the risk of placental abruption. However, it is not the only reason. The acceptable gap between the vehicles, especially in frontal collisions, helps people to control the vehicle to decrease the severity of accidents.

·       Many accidents with high severity or female fatalities were related to accidents with high speed. The authors should describe why didn’t investigate the speed above 40 Km/h.

·       According to this text “With a correctly positioned lap belt, if the belt width was doubled from 50 to 100 mm, the separated area decreased from 7.0% to 3.5%.” you can suggest executive procedures to install special pregnant seat belts for pregnant drivers.

·       The authors should provide quantitative results in both the abstract as well as conclusion section.

·       The author should describe the detail of proposed model. Therefore, they should present the flowchart of study according to the study goal.

·       The authors should use more references in 2021 and 2022.

·       The authors should provide future research and executive procedures at the end of the conclusion section. They can recommend other machine learning techniques to be incorporated into the proposed approaches to obtain more accurate results.

·       It is better authors present the conclusion bullet by bullet.

Round 2

Reviewer 2 Report

This paper is a great improvement over the previous version, but some problems need further optimization:

1Line 61“We found that over that collision speed range, the area of placental abruption increased 10-fold and accounted for approximately 20% of the total abruption area owing to pressure from the steering wheel.”Here is the result of collision numerical simulation for unconstrained pregnant drivers. What is the theoretical basis?

2、Line 172In the simulation, we applied the acceleration waveform at 40 km/h (illustrated in Figure 2) to obtain the chest and lumbar acceleration waveforms.Figure 2 is a finite element model, not an acceleration waveform, not expressed here.

3、Line 221According to the simulation results in Figure 7, when the lap belt was positioned incorrectly for the upper or middle abdomen, the abruption area increased considerablyFigure 7 is the waist acceleration waveform diagram, which is not related to the conclusion reached here.

4Line 226The rear-to-front vehicle distance is an important factor, as well as vehicle speed. When the distance is short and the preceding vehicle stops suddenly, even if the succeeding vehicle brakes as quickly as possible, the succeeding vehicle speed at collision is estimated to still be high, and the high-speed collision increases the area of placental abruptionThe situation here is rear-end collision. The previous study of this paper is mainly frontal collision, and the impact of rear-end collision on placental abruption was not studied.
